



# Refining patterns of melt with forward stratigraphic models on stable Pleistocene coastlines

Patrick Boyden[1], Paolo Stocchi[2], and Alessio Rovere[1,3]

[1]MARUM – Center for Marine Environmental Sciences, University of Bremen, Germany
[2]NIOZ Royal Netherlands Institute for Sea Research, Department of Coastal Systems (COS), and Utrecht University, The Netherlands
[3]Ca' Foscari University of Venice, Italy

**Correspondence:** P. Boyden (pboyden@marum.de)

**Abstract.** The warmest peak of the Last Interglacial (ca. 128–116 ka) is considered a process analogue, and is often studied to better understand the effects of a future warmer climate on the Earth's system. In particular, significant effort has been made to better constrain ice sheet contributions to peak Last Interglacial sea level through field observation of paleo relative sea level indicators. Along tropical coastal margins, these observations are predominantly based on fossil shallow coral reef
sequences, also thanks to the possibility of gathering reliable U-series chronological constraints. However, the preservation of many Pleistocene reef sequences is often limited to a series of discrete relative sea-level positions within the interglacial, where corals suitable for dating were preserved. This in turn, limits our ability to understand the continuous evolution of paleo relative sea-level through an entire interglacial, also affecting the possibility to unravel the existence and pattern of sub-stadial sea level oscillations. While the interpretation of lithostratigraphic and geomorphologic properties is often used to overcome
this hurdle, geological interpretation may present issues related to subjectivity when dealing with missing facies or incomplete sequences. In this study, we try to step back from a conventional approach generating a spectrum of synthetic Quaternary subtropical fringing reefs for a site in southwestern Madagascar (Indian Ocean). We use the DIONISOS forward (Beicip Franlab) stratigraphic model to build a fossil reef at this location. In each model run, we use distinct Greenland and Antarctica Ice Sheet melt scenarios produced by a coupled ANICE-SELEN glacial isostatic adjustment model. The resulting synthetic
reef sequences are then used test these melt scenarios against the stratigraphic record. We propose that this sort of stratigraphic modelling may provide further quantitative control when interpreting Last Interglacial reef sequences.

## 1   Introduction

Understanding the uncertainties surrounding the rate and magnitude of ice loss from the Greenland Ice Sheet (GrIS) and Antarctica Ice Sheet (AIS) (e.g., DeConto and Pollard, 2016; Edwards et al., 2019; Noble et al., 2020) is key to estimate
the sensitivity of our planet to warmer climatic conditions. To better constrain these uncertainties, significant effort has been made to understand how Earth's ice-sheets have evolved during past warm periods (Tierney et al., 2020). Marine Isotope Stage (MIS) 5e, here referred to as the Last Interglacial (LIG, ca. 128–116 ka) is regarded as a process analogue for future warmer climates. In the LIG, global temperatures were 1–2 °C warmer and there is general consensus that eustatic sea level was up to



7-8 meters higher than present (Dutton et al., 2015a; Dyer et al., 2021). This consensus stems from geologic work on different

types of sea-level indicators found along the world's shorelines (Rovere et al., 2016). However, peak LIG eustatic sea level estimates vary widely, also because different locations are subject to vertical land motions that need to be accounted for before reconstructing paleo global mean sea level (GMSL) from field data. These post-depositional processes include glacial isostatic adjustment (GIA), sediment loading, subsidence, tectonics, and dynamic topography (Creveling et al., 2015; Simms et al., 2013; de Gelder et al., 2020; Malatesta et al., 2022; Austermann et al., 2017).

A further challenge resides in interpreting, within fossil LIG coastal sequences, short-lived sea-level oscillations, or sudden accelerations. In the granitic Seychelles, for example, isolated in situ MIS 5e corals have been sampled and surveyed 8 m above mean sea level (a.m.s.l.) with similar facies found directly offshore in the modern intertidal (Dutton et al., 2015b). Previous lithological descriptions by Montaggioni and Hoang (1988) and a detailed modern investigation by Vyverberg et al. (2018) suggests that alternating reef sequences and coral rubble/coralline algae layers could represent a step-wise increase in sea

level through out the LIG. However, these are isolated pockets of preserved reef material and it is uncertain as to whether this sequence is derived from short-term environmental change (e.g., wave climate, coral bleaching, increased sediment input), or if in deed it is evidence of a broader series of sea-level rise and still stand events within the LIG. Off the western coast of Australia, O'Leary et al. (2013) found evidence of a slow increase in sea level before a 9 m a.m.s.l. peak at the end of the LIG. Recent reevaluation of sea-level indicators in the Bahamas show a significantly lower eustatic sea level (1.2–5.3 m, 95% confidence

interval) with an initial peak and no evidence of a late-LIG peak (Dyer et al., 2021). Empirical attempts have also been made to extract global LIG sea-Level patterns, Kopp et al. (2009) provide a statistical approach to integrate existing individual sea-level indicators into a global model based on probability densities and produced a probable 'double-peak' eustatic curve with both an early and late peak in LIG sea level. Such sudden high-to-low swing in LIG sea level cannot be identified in ice proxies (Barlow et al., 2018).

Unfortunately, solutions to this dilemma are dependent on the facies that are preserved and their geographic distribution in any given region. Unlike late Holocene studies in the tropics, where nearshore seismic and drilling campaigns on coral reef slopes can synthesize broad lithostratigraphic sequences, Pleistocene studies usually rely on either poorly preserved or spatially limited emerged reefs. Due to the inaccessibility of land based seismic and drilling on most emergent reef complexes, other methods are needed in order to supplement field observations and test interpretations.

One potential solution lies in the use of forward stratigraphic models (FSM). Over the last two decades, a suite of 2D and 3D carbonate specific FMSs have been developed either to aid in petroleum exploration (large, basin scales) or in recent Quaternary investigations (small, reef specific scale). These include CARB3D+ (Warrlich et al., 2002, 2008; Barrett and Webster, 2012), pyReef-Core and Bayesreef (Salles et al., 2018; Pall et al., 2020), ReefSAM (Barrett and Webster, 2017), and DIONISOS (Granjeon et al., 1999). While each model has individual merits, DIONISOS has proven to be consistently robust in predicting

intermediate scale reef development over the Quaternary (Seard et al., 2013; Montaggioni et al., 2015). Building on the success of Montaggioni et al. (2015) to model tropical atoll development through the Quaternary, this study aims to explore the late-LIG sea level jump conundrum through the application of a suite of GIA models to fringing reef inside DIONISOS. Here, an idealized version of the modern fringing reef system observed offshore southwestern Madagascar is subjected to subsequent





glacial/interglacial cycles starting at 400 ka before deviating from the baseline model at the start of the LIG. Resulting 3D
shorelines and synthetic well-logs aim to re-evaluate sub-stadial melting patterns previously interpreted from outcrop evidence
in Boyden et al. (2022).

## 2 Methods

This study uses DIONISOS (OpenFlow Suite, v. 2021.1, Update 4) from IFP Energies nouvelles and BeicipFranlab to synthe-
size a suite of emergent LIG coral reef terraces along the southwestern coast of Madagascar, at a site called "Lembetabe" . In
order to test the sensitivity of the geological record to changing sea-level conditions, we model three distinct relative sea-level
scenarios extracted from a coupled ANICE-SELEN model.

### 2.1 Study area

Lembetabe sits along the remote, southwestern coast of Madagascar (Figure 1). Surrounding the small fishing village, semi-
vegetated rolling dune fields back a well-preserved LIG reef sequence that sits several meters above a modern wide fringing
reef (see Boyden et al., 2022, for stratigraphic and geomorphic description). Additionally, situated on a stable coastline since
the Eocene, and in the far-field of Quaternary ice sheets, Lembetabe provides an excellent opportunity to examine more subtle
changes in eustatic sea level (Du Puy and Moat, 1996).

### 2.2 Topo-bathy generation

Unlike subsiding coastlines like the Mururoa Island atoll investigated in Montaggioni et al. (2015), Lembetabe's emergent
fringing coral reef has not been subject of extensive coring or seismic campaigns and therefore no high-resolution nearshore
bathymetry or subsurface horizons exist. In order to address this, we derive a high-resolution land DEM from aerial-based
Structure from Motion / Multi-View Stereo (SFM) (see Boyden et al., 2022, for methodology and processing description) that
have been merged down to TanDEMx topography at 30 m resolution (©DLR 2021). In order to capture the fringing reef, we
extracted the modern nearshore bathymetry at Lembetabe from WorldView 2 satellite imagery (v. 28.4, May 2020) using the
SPEAR Relative Water Depth toolset within ENVI® (v. 5.3.1). This bathymetry dataset was previously calibrated by Weil-
Accardo et al. (2022) using sonar soundings obtained with a Deeper Smart Sonar Pro™ (www.deepersonar.com) sonar unit.
These depth transects were then corrected to the established local mean sea level tidal datum from Boyden et al. (2022) before
being used for the absolute depth calibration step in the ENVI toolset. Topography, nearshore bathymetry, and the much coarser
resolution GEBCO database offshore bathymetry (https://download.gebco.net/, last access 20 November 2021) were resampled
with cubic convolution in Surfer (Golden Software, v. 21.2.192) where artifacts from the merging process were corrected and
then the entire DEM was exported to the final 50 m x 50 m model grid resolution (Fig. 2). Once exported, the decimated
DEM was imported into the DIONISOS environment where a 18 km x 5 km domain was centered at Lembetabe. This domain
was then rotated 15°from the horizontal so that the shoreline runs approximately horizontal through the gridded domain. By



doing this, intra-grid incident angles during wave propagation calculations can be minimized (e.g., Roelvink et al., 2009). Final
model domain patterns are summarized in Table 1.

## 2.3 Sediment classes

In order to resolve complex reef environments, DIONISOS utilizes user-defined sediment classes. Each sediment class is
determined by type (e.g., carbonate, clastic), grain size, density, and governing compaction law (either observed or mechanical).
In this model, we classify the seven sediment contributors, five coral specific classes that follow the same system used in Seard
et al. (2013), a standard carbonate sand class, and a standard carbonate mud class (Table 2). Initial sediment class distribution
is defined by the substratum constituents. Here, the substratum, referred to as 'Basement', is an equal distribution of the
coral sediment classes and is set to a thickness of 20 m. This is an approximation for the region as no borehole data exists.
Furthermore, to the east of Lembetabe are 30 m high limestone escarpment of Eocene age, suggesting the basement in the
region is limestone as well (Battistini, 1964; Du Puy and Moat, 1996).

As described by Boyden et al. (2022), sediment along the Lembetabe coast is dominated by carbonate with little siliciclastic
component and therefore no fluvial clastic component is included in the model. It should be noted that during glacial periods,
increases in precipitation in the hinterland could occur and an influx of clastic sediment could enter the system, but is ignored
for the goals of this investigation. While de-watering of compacting sediments in siliciclastic systems can have large effects
on the overall water budget (Revil et al., 2002), carbonate dominated systems have much more varied porosity that is subject
to significantly more alteration during the early stages of burial (i.e., diagenisis, Lee et al., 2021). The overall reef architecture
and therefore ensuing framestone lessens the changes in pore-space at shallower burial depths and therefore we treat the
reef "sediment" compaction as negligible. However, resulting sand and debris facies are governed by a simplified mechanical
compaction that is linear to depth. Finally, while DIONISOS has the ability to include fluvial input, this is also treated as
negligible as the three rivers in the region: Onilahy River (140 km to the north), the Linta River (30 km to the southeast), or the
Menarandra River (70 km to the southeast), flow sporadically throughout the year and are unlikely to have provided significant
sediment loads to the system (Fig. 1).

## 2.4 Hydrodynamics

Dominant wave direction and magnitude have significant influence on shoreline geometry as well as the ability for coral
colonies to flourish (Gischler et al., 2019). DIONISOS quantifies this impact by calculating wave energy and refraction based
on Snell's law:

$$\frac{sin\beta_d}{C_d} = \frac{sin\beta_s}{C_s} \tag{1}$$

where $\beta_d$ is the wave angle in deep water, $\beta_s$ is the wave angle in shallow water, $C_d$ is the wave velocity in deep water, and
$C_s$ is the wave velocity in shallow water (Holthuijsen, 2007, e.g.,). In order to address this, the mean observed significant
wave height ($H_s$) and direction were extracted from the Centre for Australian Weather and Climate Research (CAWCR) global





hindcast raster. This global wave hindcast was extracted from the WaveWatch III wave model between 1979–2010 in a 0.4°x
0.4°global grid. The closest grid values were extracted for the Lembetabe area.

In addition to normal sea state conditions, storms play a crucial role in coral reef development and colony long term sta-
bility (Gardner et al., 2005). While the southwestern facing shoreline of Madagascar is generally sheltered from approaching
tropical cyclones spawning in the Indian Ocean, Tropical Cyclone Haruna, a category 3-equivalent cyclone, spawned in the
Mozambique Channel and made landfall at Toliara, approximately 150 km north of the study site (Côté-Laurin et al., 2017).
Therefore, to approximate for storm impacts, the maximum significant wave height from the CAWCR model was extracted
and added to the model at a 10% yearly occurrence rate. The wave parameters are summarized in Table 3.

## 2.5 Carbonate production and facies identification

Carbonate production is naturally the largest variable in a carbonate FSM. Several factors directly control the dominate coral
species and rate of growth, in particular; water temperature, turbidity, wave energy, and water depth (e.g., Montaggioni and
Braithwaite, 2009). In order to address this, DIONISOS defines sediment classes through four parameters: (1) type of sediment
[carbonate, clastic, evaporate, etc.], (2) grain size, (3) solid density, and (4) burial compaction law [observed or mechanical].
Next, DIONISOS allows for the assignment of a a user-defined production versus depth value to each sediment class. To
streamline this process, the model's creators have provided curves for several frequently used sediment classes as defined by
literature.

For the purposes of this study, we utilize the growth curves described by Montaggioni et al. (2015) for our coral based facies,
the tabular coral growth curve from Lanteaume et al. (2018), the encrusting coral growth curve from Kolodka et al. (2016), and
the carbonate sand and mud curves from Burgess and Pollitt (2012). In the model, production is controlled as either a constant
or linear per time interval. For example, Montaggioni et al. (2015) experimented with varying the growth rates through time
and achieved significantly different results. However, because at Lembetabe we lack the seismic and well-logs available to
Montaggioni et al. (2015), our simulations are run with a maximum 2500 mMyr$^{-1}$ for coral facies, 1500 mMyr$^{-1}$ for encrusting
carbonates, 2000 mMyr$^{-1}$ for tabular corals, and 1000 mMyr$^{-1}$ for carbonate sand and carbonate mud throughout time (Fig. 3).

## 2.6 Sediment weathering

A significant limitation in the use of geological proxies to investigate intra-stadial sea-level fluctuations is a reliance on the
preservation of the target sequence. This becomes especially challenging when the target sequence is within the dynamic coastal
zone. Malatesta et al. (2022) demonstrates how overprinting can lead to ambiguities in erosional rates and inferred uplift rates of
coastal terraces. Within the DIONISOS environment, four forms of weathering are taken into account: (1) maximum weathering
rate, (2) maximum weathering decay in the marine environment, (3) dissolution rate, and (4) transformation rate. Maximum
weathering rates for exposed limestone are hard to quantify (Enos and Franseen, 1991). This is because the rate is heavily
dependent on post-depositional environmental factors, like precipitation, porosity, and groundwater chemistry. To approximate
for this, Montaggioni et al. (2015) uses a 250 mMyr$^{-1}$ that was measured by Trudgill et al. (1979) using micro-erosion meters
on Aldabra, the Seychelles. However, in the case of Lembetabe, mean annual precipitation is well below that seen on Aldabra





or Mururoa (>1000 mm vs. 62 mm) and subaerial exposure is more likely to be well below the one found on more tropical localities. Therefore, we utilize a maximum weathering of 100 mMyr$^{-1}$. To quantify the mechanical and bio erosion under

marine conditions, we use a maximum 100 m Ma$^{-1}$ weathering rate. To simplify computational load, the dissolution rate was incorporated within our maximum subaerial erosion rate. Finally, transformation rate allows for the transfer of mass from deposited carbonate into carbonate sands and muds. Within DIONISOS this is treated as a constant rate and was set to 50 mMyr$^{-1}$ for each sediment class, except carbonate sand and carbonate mud.

## 2.7   Glacial isostatic adjustment

Accommodation space is the final governing factor in reef development and morphology (Woodroffe, 2002). As sea level rises and falls, the relatively habitable window for corals moves, sometimes drastically shifting the area of reef growth up or down slope (Camoin and Webster, 2015). Within DIONISOS, relative sea level (RSL) is controlled by a user defined curve. As described before (Section 1), beyond ice-sheet mass balance, post-depositional processes can significantly affect RSL through glacial/interglacial cycles. Generally, this comprises of GIA, dynamic topography, tectonic uplift, and subsidence.

In southwestern Madagascar, the coastline is considered tectonically stable since at least the Eocene and therefore GIA was treated as the primary post-depositional influence. To take this into account we utilize solutions from a coupled ANICE-SELEN model (Bintanja and van de Wal, 2008; de Boer et al., 2013, 2014). When coupled together, the ANICE model provides the 3D ice-sheet extents for North America, Eurasia, Greenland, and Antarctica to the SELEN model at 1000 year time-steps, which then calculates the redistribution of water mass and deformation of the solid Earth (de Boer et al., 2013; Spada and Stocchi,

2007). Once calculated, SELEN sends a new RSL and deformation back to ANICE in order to solve the next time-step. The final output is a RSL curve that incorporates the GIA signal for a defined geographic location.

## 2.8   Scenarios and testing

In order to test the two prominent GrIS and AIS melt patterns within DIONISOS, we extracted three RSL scenarios from the ANICE-SELENE model for the last 410 ka (Fig. 4a). Initial DIONISOS model runs were driven by an RSL curve where

ice-sheets are held to modern geometries at the beginning of the LIG (Fig. 4b). By halting the melting of both GrIS and AIS, the resulting signal is the background GIA response. This background GIA scenario is referred to as 'Baseline' and has a peak sea level of 2.11 m a.m.s.l. at 126 ka. The second scenario, 'Full', represents a initial simultaneous collapse of GrIS and AIS, contributing 2 m and 5 m respectively (Fig. 4c). This second scenario produces an early peak of sea level of 8.33 m a.m.s.l. by 125 ka, followed by a relatively-stable, gentle regression. An early peak, driven by continental levering and ocean syphoning,

has often been seen as the classic LIG fingerprint in the far-field. In the third scenario, 'G2A5', the contributions from GrIS and AIS are separated, creating a two-peak LIG sea-level history. Here, GrIS melts first and contributes 2 m to sea-level rise between 126–124 ka that produces the initial peak of 3.55 m a.m.s.l.. Following a stable period in sea level, AIS begins to melt at 118 ka and contributes 5 m to sea-level rise late in the LIG producing a second, higher peak sea level of 6.43 m a.m.s.l. (Fig. 4d). This scenario is consistent with the interpretation of fossil reef sequences from western Australia described by O'Leary

et al. (2013).



The DIONISOS model was first run under Baseline forcing. This was done in a three-segment process where the model was run at 10000 yr time intervals between 400–150 ka. At 150 ka, the time intervals were reduced to 1000 yr intervals until 80 ka. Then, from 80–0 ka, time intervals were increased again to 5000 yr. This modulation of time intervals allowed for the reduction of computational demand while emphasizing the LIG. Following the completion of models for each mantle viscosity under Baseline conditions, the Full and G2A5 scenarios were run from the Baseline 150 ka time-step. This iterative process is summarized in Fig. 5.

## 2.9 Model caveats

As with any model environment certain limitations arise when transferring the physical environment into the virtual, often leading to an oversimplification of complex physical systems. The model used in this study is not an exception. Within the scope of this study, the primary interest is to evaluate shallow water reef evolution through glacial/interglacial cycles. DIONISOS solves sediment transport between cells using a diffusion equation (Granjeon et al., 1999). As pointed out by Barrett and Webster (2017), this simplification works well for large-scale, basin-wide models but fails to capture the more nuanced hydrodynamics within shallow reef environments. In order to address this without redesigning the entire DIONISOS architecture, characterization of the nearshore wave environment allows for a reasonable approximation at our model grid resolution of 50 m x 50 m.

The largest caveat however, is the method in which DIONISOS solves for subaerial erosion. This is partially addressed through the use of user-defined weathering rates for each sediment class. However, in Lembetabe, modern active dune fields and fossilized aeolianites cover MIS 5e reefs, suggesting extensive offshore migration of dunes during glacial periods (e.g., Battistini, 1965; Boyden et al., 2022). The speed at which active dunes migrate, would limit the overall exposure time of previously deposited limestone (Bristow et al., 2005). In order to try and limit the over estimation of subaerial weathering, we employ a lower weathering rate than regularly cited for coastal limestone facies (e.g. 100 vs. 250 mMyr$^{-1}$, Trudgill et al., 1979).

## 3 Results

The results from the DIONISOS model are organized two fold in the following section. First, transgression and regression timings for each scenario are described. Second, in order to better visualize the differences between the sea-level scenarios adopted, three synthetic wells were created along a cross-platform profile within the baseline model output. The first well (A) is located just before the continental-shelf break, the second (B) is within the reef, and the third (C) is located in the intertidal. Well location can be seen in Fig. 6.

## 3.1 Facies Deposition

Within the post-processing model environment, seven facies were declared following the coral distribution scheme laid out by Montaggioni (2005) for reef dominated coasts in the Indo-Pacific basin as well as model specific environmental constraints





from Seard et al. (2013). The relationships between sediments and environmental controls within the model output are summarized in Fig. 7. Each facies is a combination of one or more of the sediment classes described before (Section 2.3) and deposition depth. Further classification parameters are possible but are beyond the scope of this study and require significant

fine-tuning. Moving from off-shore over the reef, we declared: (1) Forereef, (2) Reef crest, (3) Reef-flat, (4) By Pass, (5) Beach, (6) Basement, and (7) Unclassified. Applying the declared facies to each scenario, the following depositional histories were obtained for each scenario at each well location.

### 3.1.1 Baseline

Well A, sitting near the edge of the platform, first registered transgression at 131 ka with the initiation of a thin, 2 m thick

reef crest. Sea level continued to rise with the back-stepping of the reef crest and subsequent deposition of a forereef sequence between 130–115 ka. Sea level began to fall once again and hydrodynamic conditions allowed for the creation of a second 4 m thick reef crest sequence to be deposited. By 110 ka, sea level had regressed back beneath the platform edge, leaving the LIG exposed and subject to weathering. Burial analysis of the sequence shows that the upper 7.5 m of LIG limestone to have eroded away leaving the 116 ka forereef unit the highest remaining LIG unit in the resulting lithostratigraphy (Fig. 8a). At 15

ka there is again transgression of the platform and rapid deposition of a 13.9 m thick sequence of forereef by 5 ka.

At Well B, midway on the platform, the Baseline model shows a rapid flooding starting at 130 ka. Between 129–122 ka, a 23.3 m thick sequence of reef crest was deposited. This was followed by a 3 m sequence of an unclassified facies, most likely a reef-flat-like sequence that did not meet the defined parameters stated before (Fig. 7). The unclassified sequence was capped-off by a thin beach deposit by 120 ka. This beach deposit was a thin veneer that reached no more than 0.5 m thick and

covered the LIG reef crest sequence from 120–88 ka. Subaerial erosion began taking place, eroding 3.5 m from the uppermost part of theLIG sequence by 10 ka. At that point sea level transgressed again, depositing a thin 1.5m-thick Holocene reef crest that was subsequently covered by a equally thin Holocene reef-flat deposit.

Well C occupies the modern intertidal. The baseline LIG sea level transgression reached the well at 127 ka. The relative stable nature of the background GIA signal at Lembetabe created a prograding fringing reef. Here, a 1m-thick sequence of reef crest

was deposited between 127–126 ka. This reef crest sequence was followed by a reef-flat sequence 2.5 m thick between 126–125 ka. As sea level continued to recede, a 1 m thick sequence of alternating beach and unclassified units were deposited. By 124 ka, the sequence was left exposed and the upper 2.5 m of the sequence were eroded away before the Holocene transgression at 10 ka. The Holocene transgression deposited further reef-flat units on top of the LIG up to modern sea level.

### 3.1.2 Full

The platform edge under the combined GrIS and AIS, 'Full' scenario, is initially flooded at 131 ka. A quick initial pulse of forereef deposition occurs between 130–125 ka, resulting in a 9 m thick unit. Deposition slows up until peak sea level is reached at 120 ka, after which sedimentation rates increase as sea level falls again. This leaves behind a 3 m thick unit of forereef between 120–113 ka. As sea level continues to fall, a 3.5 m reef crest unit is deposited in the remaining accommodation space by 111 ka, before the platform becomes emerged and a thin 0.5 m beach unit is deposited on top. The uppermost 10 m



of the LIG are then eroded away during the intervening subaerial exposure. At 15 ka, the Holocene transgression reoccupies the platform and 18 m of forereef are deposited on top of the remaining LIG from 15–5 ka. Following 5 ka, sea level begins to slightly recede, allowing for the deposition of a 15 m thick reef crest unit.

Similar to the platform edge, transgression during the LIG begins by 130 ka with an initial 0.5 m thick reef crest unit. This is then quickly covered by forereef units interbedded with unclassified sediments as sea level continues to rise until 125 ka. 255 In total, 25.3 m of forereef and unclassified sediments accumulate before sea level begins to fall. Between 125–119 ka, an additional 1 m of fore reef sediment accumulates before conditions be conducive for the deposition of 2.9 m of reef crest. Sea level drops yet further and a final phase of sedimentation sees 1.2 m of reef-flat before a final veneer of beach is deposited by 116 ka. The section becomes emerged and undergoes subaerial erosion until 10 ka, eliminating the upper 8.5 m of the LIG sequence. The final Holocene sequence comprises of 1.4 m of reef-flat topped with shallow-water unclassified sediment that 260 reaches present day sea level.

The LIG for Well C under the full scenario produces an initial inundation at 127 ka when a 10 cm thick unit of reef crest was deposited. This is followed by additional sea-level rise and the deposition of a 3.5 m thick forereef unit by 125 ka. Following this, accommodation space is decreased and a 4.4 m thick reef crest unit is deposited. Further shallowing and progradation cause the deposition of a final 1.8 m thick sequence of reef flat and 0.8 m thick shallow-water unclassified sediment unit by 265 122 ka. This was left subaerially exposed and the uppermost 3.5 m of the LIG were eroded, leaving the remaining LIG reef crest exposed at +3.5 m a.m.s.l. (Fig. 8c).

### 3.1.3 G2A5

Transgression over the platform edge begins at 132 ka under the G2A5 scenario with the deposition of a 1 m thick reef crest. Rapid sea-level rise causes back-stepping of the reef made evident by the deposition of a 9 m thick reef crest occurs between 270 131–125 ka coinciding with the GrIS melt contribution. At 125, RSL begins to decrease, eroding the upper 1.5 m of the earlier deposited reef crest by 119 ka. AIS melting influence is then gradually felt, again slowly depositing a reef crest unit between 118–113 ka. By 113 ka, rapid increase in sea level is seen and 2.5 m of further forereef are deposited by 110 ka. As sea level regressed, a 5.6 m thick sequence of reef-flat and unclassified units were deposited between 110–109 ka. Finally, the LIG sequence was covered by interbedded beach and unclassified deposits reaching a maximum of 8.8 m thick, as sea level fell 275 back beyond the edge of the platform.

At Well B the LIG sequence begins at 130 ka with the deposition of a 0.5 m thick reef crest unit. This is then covered by the rapid deposition of 33.3 m of reef-flat from 127–116 ka, again coinciding with GrIS melt contributions. It should be noted that this peak deposition is over 5 m modern a.m.s.l.. Following the initial peak in sea level, Well B is left exposed and 17.1 m of previously deposited reef-flat is eroded by 80 ka. Erosion continues until the onset of the Holocene, when 10.3 m of reef crest 280 is deposited by 5 ka and a final 3 m thick unit of reef-flat is deposited up until modern.

Finally, Well C records the initial LIG transgression at 126 ka. This comprises of a 4.6 m deposit of reef-flat by 124 ka. Above the reef-flat, a 0.5 m thick beach unit is deposited by 123 ka, but is subsequently partially eroded during the intra-peak regression. AIS driven sea-level rise re-inundates the intertidal and deposits a 2.8 m thick unit of reef-flat between 119–117



ka. A final 0.8 m thick beach unit is deposited atop the LIG sequence by 116 ka. As at Well B, rapid regression occurs and the
entirety of the LIG sequence is eroded away. Reoccupation of the platform at 10 ka, deposits a final 2 m thick unit of reef-flat
that is visible in the final litholog (Fig. 8c).

## 4   Discussion

This implementation of DIONISOS focuses on two commonly cited scenarios for GrIS and AIS melt contributions to the
LIG by comparing the fringing reef accretion under the respective GIA-derived RSL curves against a reef sequence created
using a pure background GIA RSL history. Our results show distinct differences in nearshore reef accretion during the LIG
as well as stark differences in the preservation of LIG in the modern stratigraphy under different ice-sheet melt scenarios.
In the following we: (1) quantify the differences in accretion, (2) examine preservation shortcomings, and (3) evaluate the
applicability in real-world LIG fossil reef based studies.

The two main direct consequences of changes in ice-sheet melting parameters are changes in the amplitude and duration
of maximum inundation. This has direct influence on the overall accommodation space available for reef development (e.g.,
Camoin and Webster, 2015). At Lembetabe, the combined GrIS and AIS melt under the Full scenario produces a 295% increase
in RSL when compared to the Baseline. Under the G2A5 scenario this difference is marginally less at 205%, when comparing
the highest experienced sea level. This translates to LIG water depths over the mid-platform (Well B) reaching a maximum
of 17 m under Baseline, 18 m under G2A5, and 22 m under the Full scenario. The rapid flooding of the reef platform at the
beginning of the LIG, would have placed the antecedent topography directly in the ideal zone of coral growth (Woodroffe
and Webster, 2014). Sedimentation rates across theLIG from the mid-platform reflect this as well with rates for all scenarios
peaking at around 3600 mMyr$^{-1}$. Within the stratigraphy, this is recorded as both vertical accretion as well as progradation as
accommodation space decreases driven by both RSL fall as well as reef growth (e.g., 100 ka in Fig. 6b).

By the end of the LIG, a thick carbonate package covered the platform. Under baseline conditions, maximum LIG thickness
reaches 24.5 m, under the Full scenario the maximum thickness decreases to 24.1 m, and under G2A5 it decreases further
to only 13.0 m. However, the higher overall RSL rise during the two melting scenarios, spread accretion across the entire
platform and inland. This significant increase in inundated area lead to a 681.5% increase in carbonate sedimentation under
the Full scenario when compared to the Baseline, and a 187.5% increase under the G2A5 scenario compared to the Baseline.
The spatial increase of inundation is highlighted in Fig. 9. Furthermore, comparison between the scenarios at Well C, show a
significant difference in facies occurrence. Under the Baseline model, only a small amount of LIG reef is preserved above the
basement (Fig. 8a) and under the G2A5 scenario, there is no LIG reef present (Fig. 8c). This narrative is reversed under the
Full scenario where the LIG reef comprises of an approximately 3 m section of forereef, capped off by an approximately 2 m
section of reef crest (Fig. 8b). The limited record of the LIG in the other two scenarios is most likely a reflection of one of
two factors; (1) the lower overall RSL would have placed the active reef crest further seaward than Well C or (2) the shorter
overall duration of inundation during the second peak of RSL rise under G2A5 conditions would not be enough to accrete
significant reef sequences. For example, at the highest sedimentation rate of 3600 mMyr$^{-1}$, the second peak under the G2A5



melt scenario lasts for 2000 yr. This would equate to a potential 7.5 m maximum accumulation of carbonate, significantly less that the potential accumulation under the Full scenario of 32.4 m.

As described earlier in Section 2.9, erosion and sediment transport within DIONISOS is reliant on user-defined maximum values and simplified diffusion equations, respectively. While we do see preserved MIS 5e sequences within produced synthetic well logs, actual exposed MIS 5e facies are lacking. This 'preservation bias' is especially apparent when following the progression of sedimentation through the LIG. As pointed out in Section 3.1, under each scenario, multi-meter sections of lithology are removed in the intervening millennia between the LIG and modern. This is further reinforced by the exposure times under each scenarios for Well B. Under all three scenarios, shore-ward of the mid-platform is left exposed for 115+ kyr before the final phase of Holocene sedimentation (Fig. 4). Such a long exposure time would, if maximum weathering rates were maintained, lead to a potential 11.5 m of erosion.

Unlike the previous investigations (e.g., Montaggioni et al., 2015), a one-to-one comparison between the model and the fossil record is not possible. However, trends within the model output are roughly analogous to observations in the field. In particular, the presence of significant reef facies above modern sea level under the Full scenario is in good agreement with recent descriptions of the LIG facies present at Lembetabe (Fig. 10, Boyden et al., 2022; Weil-Accardo et al., 2022). Overall discrepancies in elevations of such facies are most likely derived from differences in spatial geometry of the pre-LIG coast, differences in weathering rates during glacial times, as well as possible gradual subsidence in the region. Furthermore, the geomorphology of the modern coastline and the modern fringing reef is roughly reproduced (Fig. 6).

## 5 Conclusions

In tropical and sub-tropical regions, field-evidence based constraints of LIG eustatic sea level often rely upon incomplete, temporally limited preservation of fossil coral reef facies. This, in turn, leaves the derivation of LIG sea level patterns up to interpretation and 'expert judgement'. Here, we apply a suite of GIA models to a fringing reef system offshore southwestern Madagascar within the DIONISOS forward stratigraphic model environment. The resulting stratigraphic sequences of two distinct GrIS and AIS melting scenarios during the LIG are then evaluated against the observed lithostratigraphy and a baseline, pure-GIA signal. By creating fictitious stratigraphic sequences based on physical principles, we are able to conclude that: (1) changes in the timing of GrIS and AIS melting produce substantial, visible changes in the stratigraphic record, (2) preservation within the coastal environment heavily influences the interpretation of the fossil reef and the derived paleo relative sea level, and (3) adoption of this methodology, while not a one-to-one comparison, would allow for a more objective evaluation of outcrop interpretation. We further conclude that based on these initial results, there is a compelling need for stability within the LIG, which is underlined by the lack of reef facies preserved under a two-peak sea level scenario. While significant sedimentation takes place during the LIG under a two-peak scenario, the intermittent sub-aerial exposure of the freshly deposited reef drastically decreases the robustness of the LIG record, bringing into question the possibility to observe such relative sea level fluctuations in the fossil record.





*Data availability.* Forward stratigraphic model input files and GIA model outputs are available at: https://zenodo.org/record/7565917 (Version 1.0, Boyden et al. (2023))

*Author contributions.* PB, PS, and AR conceived the project; PS provided the GIA sea level scenario inputs; PB and AR evaluated model inputs; PB ran and analyzed model runs; PB wrote the initial manuscript; all authors reviewed and edited the manuscript.

*Competing interests.* The authors have no competing interests to declare.

*Acknowledgements.* This research was funded by the DFG (Deutsche Forschungsgemeinschaft) Excellence Cluster "EXC 2077: The Ocean Floor – Earth's Uncharted Interface" (Project number: 390741603), the European Research Council (ERC) under the European Union's Horizon 2020 research and innovation program (grant agreement No. 802414).



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



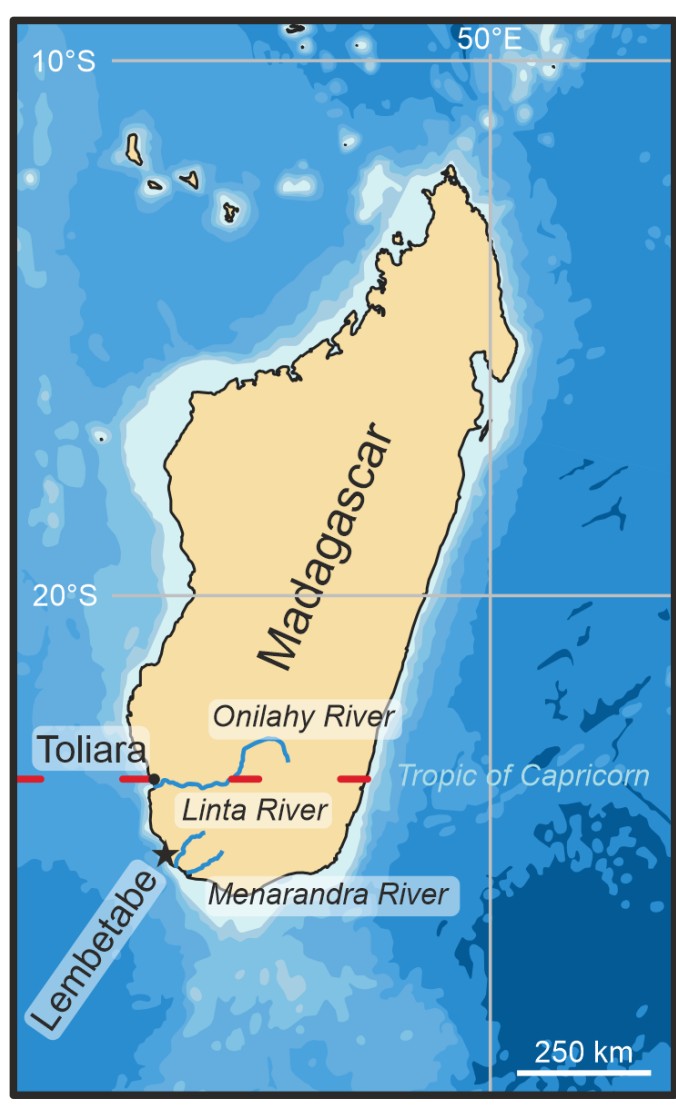

**Figure 1.** Overview map of Madagascar, the study area at Lembetabe and the three rivers in the southwest of the island. Map data from Natural Earth.



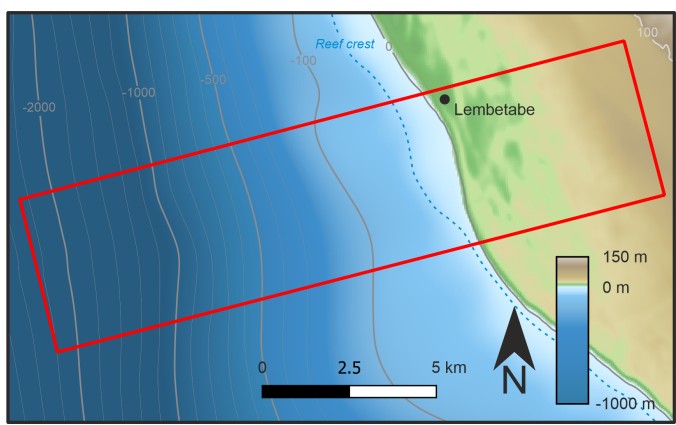

**Figure 2.** Overview of DIONISOS model domain with combined topography/bathymetry detailed in Sect. 2.2.



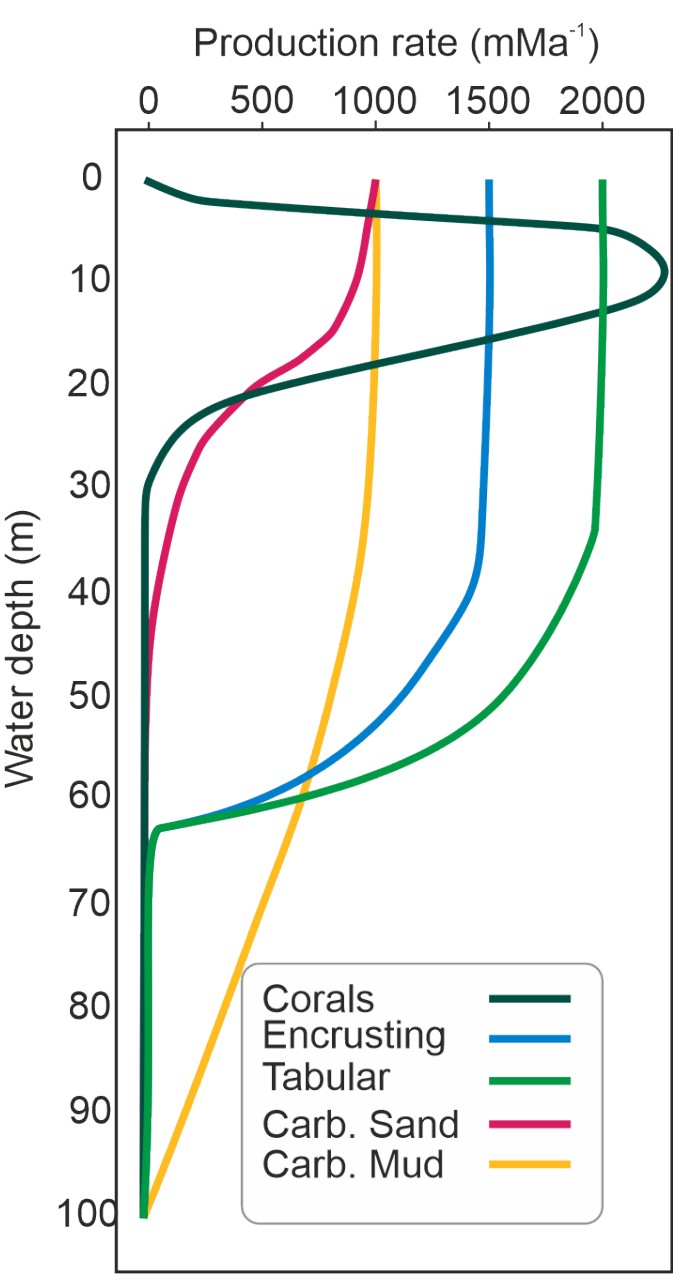

**Figure 3.** Production rate (mMyr⁻¹) as a function of water depth for each sediment class included in the DIONISOS model.



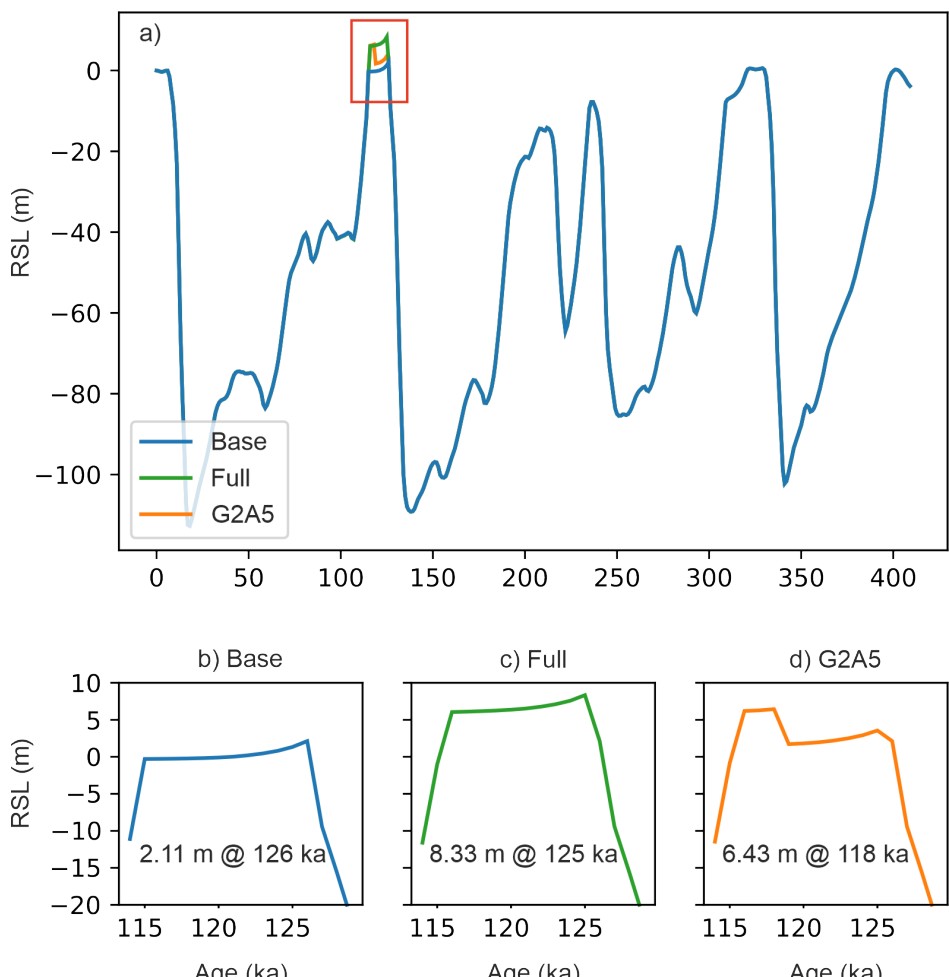

**Figure 4.** RSL sea level curves extracted from ANICE-SELEN for Lembetabe and used in DIONISOS. (a) Overview of RSL from 400–0 ka. (b) Baseline curve with no further melting of GrIS or AIS beyond modern geometries at LIG. (c) Full RSL curve, with joint melting of GrIS (2 m) and AIS (5 m) at the onset of the LIG. (d) G2A5 RSL curve, with initial 2 m contribution from GrIS, followed by a later contribution of 5 m by AIS. Magnitude and timing of peak RSL in each scenario is included for reference.



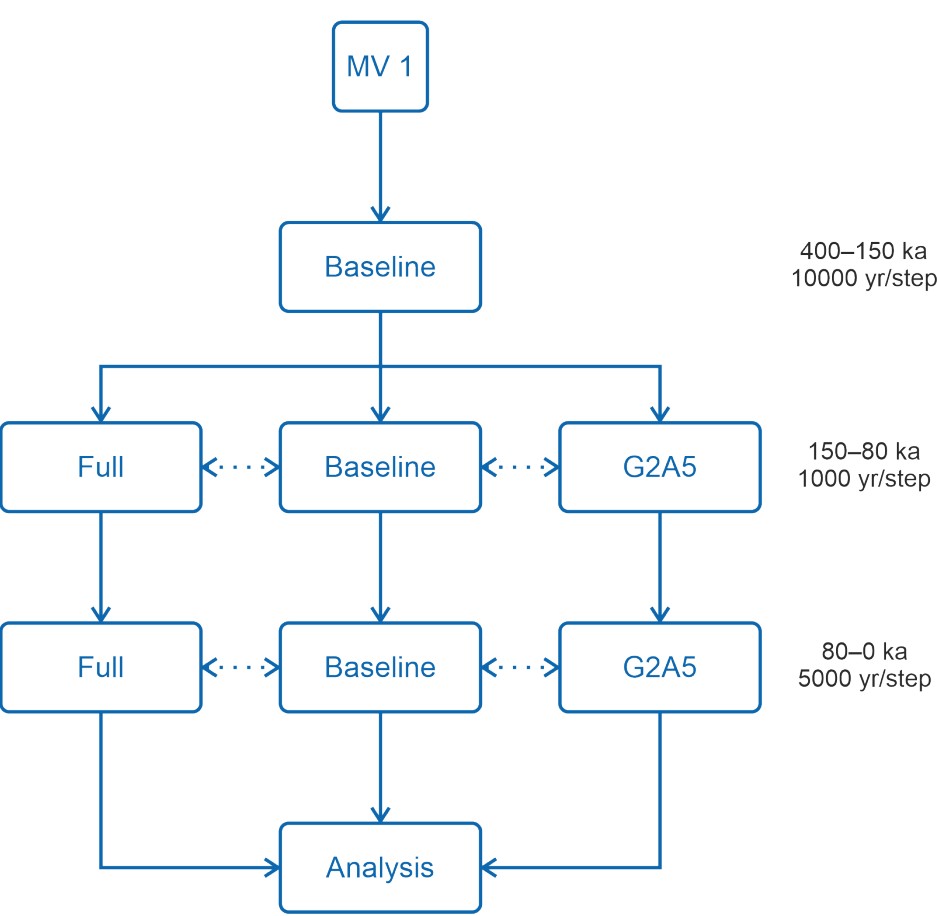

**Figure 5.** Outline of experimental design. MV 1 represents the mantle-viscosity scheme used during the ANICE-SELEN model calculations. 'Full' represents an early LIG peak in sea level that is driven by simultaneous melting of GrIS and AIS, contributing 2 m and 5 m respectively. 'Baseline' is the background GIA signal during the LIG, where ice-sheets are kept to modern geometries throughout the LIG. 'G2A5' is a two-step LIG scenario where GrIS melt adds 2 m to sea level at 125 ka followed by a brief plateau before AIS melt contributes 5 m at 118 ka.





**Figure 6.** DIONISOS model output for major event markers: 130 ka, 125 ka, 100 ka, and modern. (a) Baseline model driven by background GIA, (b) Full model incorporating simultaneous melting of GrIS and AIS, and (c) G2A5 model subjected to initial GrIS melting followed by AIS melting near the end of the LIG.



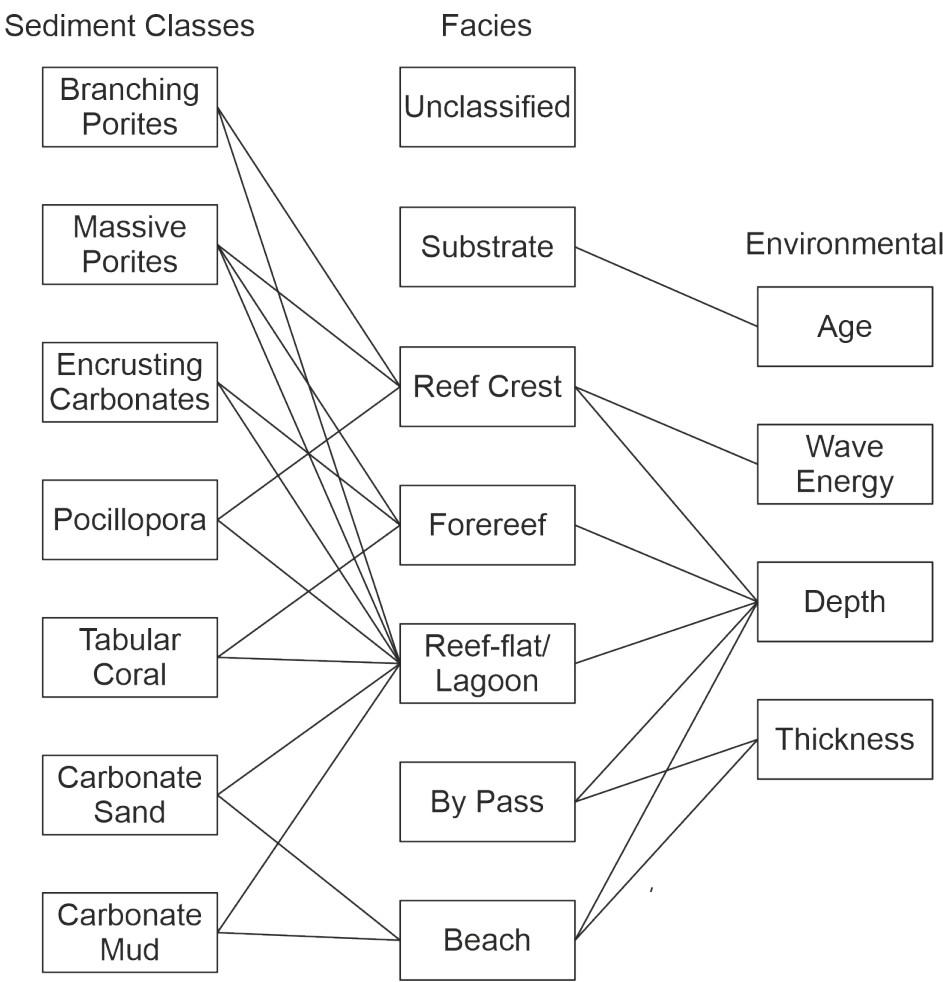

**Figure 7.** Identification pathways for determining facies within DIONISOS. Adapted from Montaggioni (2005) and Seard et al. (2013).







**Figure 8.** DIONISOS model well logs for: (a) Well A, (b) Well B, and (c) Well C. Water depth is depth of original deposition and Age is time of deposition. Line color corresponds to scenario, orange = Base, blue = Full, and green = G2A5.



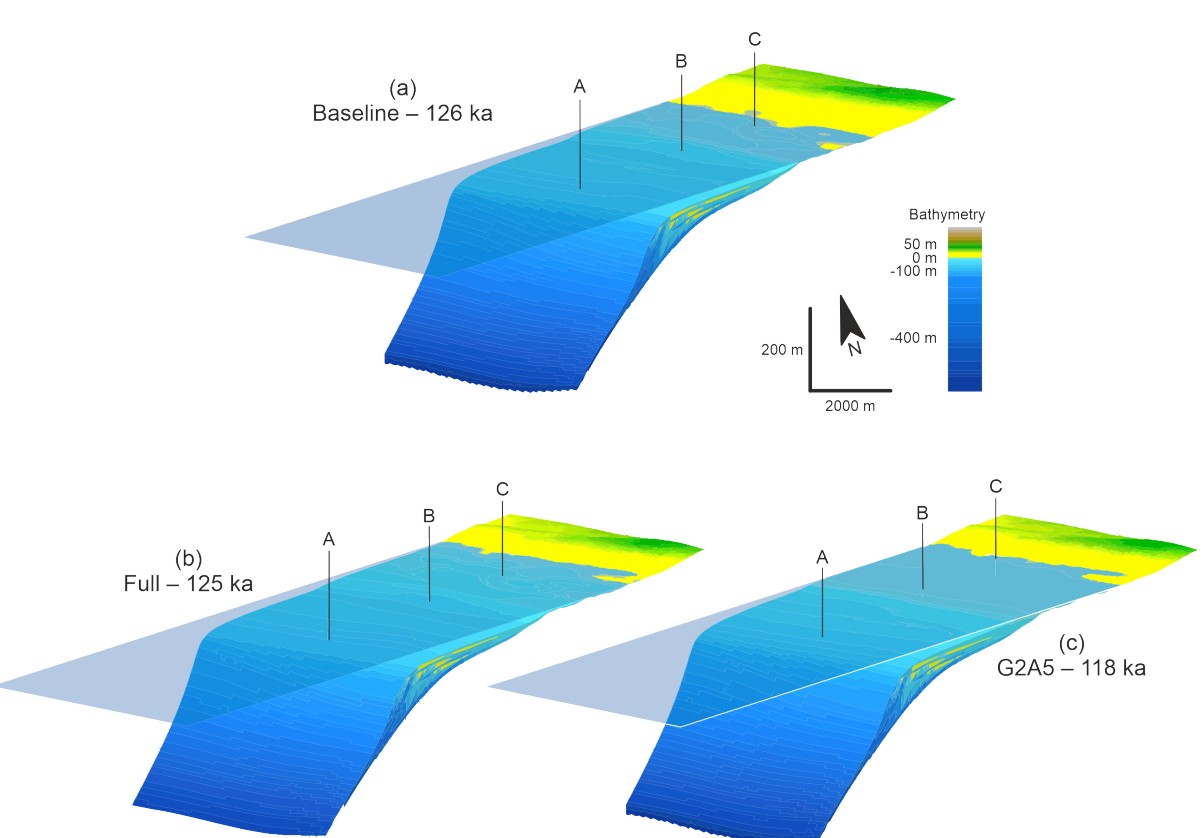

**Figure 9.** Maximum inundation time-step for each scenario. (a) Baseline model run with peak sea level at 2.11 m a.m.s.l. (b) Full scenario with peak sea level at 8.33 m a.m.s.l. (c) G2A5 scenario with peak sea level at 6.43 m a.m.s.l.

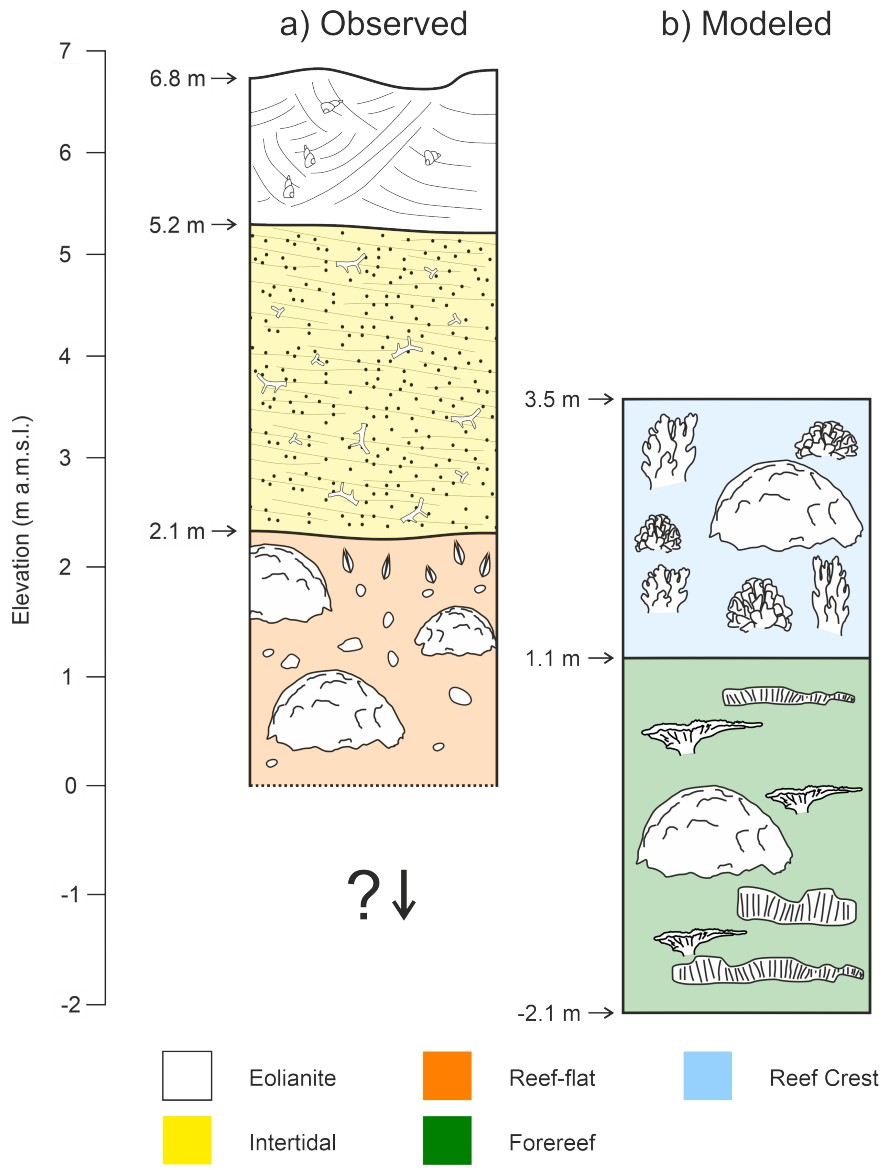

**Figure 10.** Comparison between field observations made by Boyden et al. (2022) and lithology extracted from Full scenario at Well C. (a) Synthesized litholog based on mean maximum facies elevations reported in Boyden et al. (2022). According to Boyden et al. (2022), the reef-flat correlates to the MIS 5e highstand, the intertidal facies represents the close of MIS 5e and the eolianite closes out the sequence correlating to much younger deposition. (b) Lithology at Well C for Full Scenario. Colors represent the declared lithologies within the model environment and major bioconstructors are included for cross-comparison purposes.



**Table 1.** DIONISOS model domain grid parameters used

| Parameter | Value |
| --- | --- |
| Grid length (km) | 18 |
| Grid width (km) | 5 |
| Grid resolution, Dx (m) | 50 |
| Grid resolution, Dy (m) | 50 |
| Cell Count, Nx | 360 |
| Cell Count, Ny | 100 |



**Table 2.** Sediment classes used within DIONISOS. Coral related class parameters are the same used in Seard et al. (2013).

| Sediment Class | Type | Grain size (mm) | Solid density (kg/m$^3$) | Wave energy range (kW/m) |
|---|---|---|---|---|
| Branching Porites | Carbonate | 0 | 2500 | 10–max |
| Massive Porites | Carbonate | 0 | 2500 | min–5 |
| Pocillopora | Carbonate | 0 | 2500 | 10–max |
| Tabular Coral | Carbonate | 0 | 2500 | 5–15 |
| Encrusting Carbonates | Carbonate | 0 | 2500 | 10–max |
| Carbonate Mud | Carbonate | 0.004 | 2500 | min–1 |
| Carbonate Sand | Carbonate | 0.4 | 2500 | min–5 |



**Table 3.** Model Hydrodynamic boundary conditions.

| Parameter | Value |
|---|---|
| **Mean** | |
| Wave base (m) | 6.0 |
| Propagation azimuth (°) | 30 |
| Wave energy flux (kWm$^{-1}$) | 14.02 |
| Wave height (m) | 2.15 |
| **Storm** | |
| Wave base (m) | 19 |
| Propagation azimuth (°) | 30 |
| Wave energy flux (kWm$^{-1}$) | 19.90 |
| Wave height (m) | 6.82 |