# Peer review of "Refining patterns of melt with forward stratigraphic models on stable Pleistocene coastlines"

_EGUsphere, 2023_

## Author Response (AR1)

**Response to Reviewer Comments for "Refining patterns of melt with forward stratigraphic models on stable Pleistocene coastlines"**

We would like to thank both Georgia Grant and Gino de Gelder for their insightful assessment of our work, and constructive comments for the improvement of this manuscript. Both reviewers correctly pointed out the need to provide additional discussion on both the comparison between model lithologies and those observed as well as implications of our findings on the broader interpretation of other LIG fossil shorelines. In order to rectify this, we expanded the end of the discussion section to include further comparisons of each scenario at Well C, along with further, more in depth discussion of physical limitations precluding a 'double-peak' at Lembetabe, southwest Madagascar. Finally, we added an additional paragraph to the end of the discussion indicating needed next steps to be taken with this methodology.

**Line and Figure Comments**

Individual responses to Line comments are interspaced in the following. Straightforward edits were agreed to and carried out, see manuscript.

**G. Grant**

Line 59: A brief explanation why Madagascar was chosen would be useful here, particularly as there is very little data available for the site. Is it selected primarily because it is considered stable? Explanation is found in Section 2.1 Study area.

**Line 121: Appendix table including the exact sites (lon/lats) used**

Changed sentence to read, "The resulting values from the grid cell overlaying Lembetabe were then extracted as hydrodynamic boundary conditions for the model."

Line 131: repeats from Line 92 – although written more clearly here. Remove from one location. The definition from Line 92 was replaced by the definition from Line 131 and Line 131 was then modified to:

"In order to address this, DIONISOS classifies each carbonate producer under the four component sediment class definition from Section 2.3 and then adds a user-defined production versus depth value to each sediment class."

**Line 133: I'm unclear what production versus depth curves are, sediment deposition/erosion? Please clarify.**

To clarify we added, "his production versus depth curve represents the relationship nonlinear relationship between carbonate producer growth rate and depth."

Line 141-142: how do these parameters compare to Montaggioni? Are they taken from Montaggioni or different and if different, what was the reasoning. I'm unclear on why these values were used.

To clarify, we changed the text to, "...with constant maximum growth rates from the literature mentioned above; e.g. 2500 ..."

**Line 154: repeats "Therefore, we utilize a maximum weathering of 100 m.Myr-1." On line 155: "we use a maximum 100 m Ma-1 weathering rate" and contains different units.**

The units were fixed and the sentence wording was changed to read, "...utilize a maximum subaerial weathering of 100 m Myr..." on line I54

Line 154: Reasoning for weathering rates. While I feel the maximum weathering rate is acceptable given the comparison and evidence Montaggioni and Trudgill, I think this could be explained more clearly. Was there a ref for mechanical and bio erosion under marine conditions?

In order to clarify, we added, "... weathering rate (e.g., between the lower and the average values used by Paulay et al., 1990, 0 and 1000 m  $Myr^{-1}$ )."

Line 157: Restate to make clear exactly what is referred to, there are a lot of rates and curves, these need to be explicitly stated. Here "the transformation rate is treated as a constant..." And Line 141: "Our simulations are run with a maximum x for coral facies" – add maximum growth rates. Check elsewhere.

We re-worded the text to, "Finally, transformation rate allows for the conversion of mass from deposited carbonate into carbonate sands and muds."

Line 162: for additional understanding by the reader, I suggest you add something akin to "down slope, determined by the slope steepness (ref)".

There is a misunderstanding, to clarify the text was changed to, "...reef growth either up or down the coastal profile"

Line 162: add reference to Section 2.8 with "user defined curve". While it's the next paragraph, the sentence as it stands makes you expect a description of the RSL curve, which is not forthcoming, and that makes you question whether in fact it is provided by the user or not.

To simplify, the sentence was moved to Section 2.8 as the second sentence.

Line 174: add to clarify that the model runs over the last 410 kyr, with a (global?) sea-level curve used for baseline, and the two altered scenarios only change sea-level from (130-115ka?). That you run to present to represent all post depositional processes so that the synthetic wells would be comparable to modern observations is very understated here! And a key requirement for this approach.

Clarified by adding, "...model for the last 410 ka: one baseline scenario and two scenarios where RSL changes between 130--115 ka..."

Line 321: I do not understand what this referring to "actual exposed MIS 5e facies are lacking" – do you mean in the synthetic logs? If so, reword. The terms 'actual' and 'exposed' made me initially think of real exposed outcrop. If you are referring to real outcrop then reference.

This was switched and has been corrected to read, "While we do see preserved exposed fossil MIS 5e reef facies at Lembetabe (e.g., Battistini et al., 1965 and Boyden et al., 2022), the corresponding MIS 5e sequences within produced synthetic well logs are sometimes lacking (Fig. 9c)."

Line 332-333 : is a general simplification of facies formed and classified by the model an issue here too?

No, while the uniqueness of the erosion rates at Lembetabe differentiate it from other studies, the growth rates and curves utilized here follow established Indo-Pacific basin wide values and represent the most prolific contributors to reef construction.

Line 344: I think this is somewhat overstated for the conclusions given the short paragraph discussing this at the end of the discussion. It should either be heavily edited (as below) or removed from the conclusions to discussion, or requires more development of this concept in the discussion to support this conclusion. From my understanding "We further conclude that a stable LIG sea-level signal is most consistent with the synthetic stratigraphy presented here, specifically because no reef facies are preserved under the double-peak sea-level signal." As the statement stands, you are taking one sites, for which you have limited observations and interpreting this as representative for global sea-level (which you have not previously stated). In general, I think more discussion is required regarding Fig. 10 and the facies difference and elevation differences, in fact the caption contains more pertinent information (Boydens facies interpretation of peak sea-level) that is not contained in the main text. When you call it roughly anagolous to the observation, this is not explained what the similarities ad differences are.

Lastly, I would consider including some broader implications for this type of study. Do you consider that this model is easily applied to other areas, is the baseline study and lack of GIA at this site critical? Are the differences in the resulting synthetic stratigraphy significant enough to be able to conclusively determine the LIG melt history (double peak or not). Are the different locations

of LIG sea-level expected to be so different this will need to be applied to each? What are the intended next steps/ areas that need further development?

We addressed this comment above as it was also suggested by Gino de Gelder.

G. de Gelder

Line 43: What does high-to-low swing mean here? Shouldn't it be a high-to-low-to-high swing if there's a double peak?

In order to skip the confusion, the phrasing was changed to "Such sudden dynamism in LIG sea level..."

Line 57: This sentence is a little confusing, since you start talking about the modern reef system and then about subsequent cycles starting at 400 ka. I think I know what you mean (starting with a 400 ka fringing reef based on the modern reef system), but better clarify.

Changed wording to, "Here, we take an idealized version of the modern fringing reef system observed offshore southwestern Madagascar and subject it to glacial/interglacial cycles..."

Line 113: is there wave erosion in the model as well? If yes, in what sense? If no, would this affect the outcomes?

Yes, see Section 2.6 for weathering and erosion control.

Line 140: what are all these numbers based on? If it's the same studies as for the growth curves it could be specified.

To clarify, we changed the text to, "...with constant maximum growth rates from the literature mentioned above; e.g. 2500 ..."

Line 147: You give the Malatesta example of (wave) erosion rate, but I guess that's not the same as the four weathering processes you mention here? If so it's a little misleading to cite Malatesta here.

This was cited to demonstrate the important role that weathering can play in shaping the geological record. To make it clearer, we have added, "For example, Malatesta et al., 2022..."

Line 156: Not sure if I understand the transformation rate. Does it both transfer (transfer of mass from one part of your model to the other) and transform (change carbonate into carbonate sands and muds)? Which part of the sequence is transformed, and if it's transported, to where? Please clarify

We changed "allows for the transfer of mass..." to "allows for the conversion of mass..."

Line 164: I would change tectonic uplift to tectonics, since tectonics can both result in uplift and subsidence. And what do you mean with subsidence? I would either say "anthropogenic subsidence" (to emphasize this is not tectonic subsidence) or "other vertical land motion processes" (more generic, includes anthropogenic subsidence).

Agreed, text has been changed to, "...tectonics, and other vertical land motion processes."

Line 181: Is there a scientific reason why the GrIS would be melting before the AIS, other than that it is a convenient way to create two peaks? I'm no expert, but as I understand from Govin et al. (2011) and Stone et al. (2016) it could be an explanation for different temperatures in Antarctica and Greenland during the early interglacial. Maybe mention this somewhere?

Agreed, to elaborate, we have added, "This scenario represents possible discrepancies in GrIS-AIS stability, variable hemisphere-specific climate fluctuations (e.g., Govin et al. 2012, Stone et al. 2016) and is..."

Line 296: I don't think you can really say "a 295% increase in RSL", I would rather use the amount in m. For example: when talking melt scenarios the difference between 1 and 5 m RSL is for me about as important as between 2 and 6 m RSL, yet one would be 500% and the other 300%.

The values for each peak RSL are stated in Section 2.8 "Scenarios and testing". In order to clarify this better we have included each corresponding value.

Line 306: Would be interesting to comment why the LIG carbonate package is so much thinner for the G2A5 scenario, for me the reason is not directly obvious

To further elaborate on this point, the following text was added:

"The difference in carbonate package thickness between the Full and G2A5 scenarios is driven by the duration, and corresponding depth, of maximum inundation. This is most visible at Well C, where a 5 kyr (127--122 ka) continuous occupation of the platform deposits a transgressive to prograding sequence of reef facies (reef crest--forereef--reef crest) under the Full scenario. This is in contrast to the shallower reef-flat facies that are deposited during intermittent occupations under the G2A5 scenario, 126--124 ka and 119--117 ka."

Line 307: Good to mention the amount of carbonate sedimentation as well (in m I suppose?), at least for the Baseline, just to know what kind of amounts we're talking about.

In our opinion, the overall volume of is not as important to show as the order of magnitude change between scenarios.

Line 310: This is a little confusing, the previous sentence suggests you'll talk about Well C, but Fig. 8a is about Well A. In this sentence and the next, do you mean (Fig. 8c-left), (Fig. 8c-right) and (Fig. 8c-middle) instead of Fig. 8a, Fig. 8c and Fig. 8b? Also, in this sentence you mention the basement, but I don't see the basement in Fig. 8c?

The text has been altered to clarify which sub-figure is being referenced.

Line 318: Maybe add the duration of the peak for the Full scenario as well.

Have added, "...potential accumulation of 32.4 m during the 9000 yr peak of the Full scenario."

Line 329: I think the word 'significant' is very vague here, maybe replace it with something more concrete

Added, "...significant reef facies 3.5 m above modern..." to clarify.

Line 332: I don't understand this last sentence, why 'furthermore'?

We have moved this sentence further up in the paragraph and have combined it to read: "Out of the three scenarios tested, only the Full scenario has preserved LIG reef facies above modern sea level (forereef and reef crest, Fig. 9) as well as the ability to reproduce the general geomorphology of the modern coastline and the modern fringing reef (Fig. 6)

Fig. 6: It is very difficult to see the differences between the three models. Given the huge amount of white space in each figure, I think you can find a way to increase visibility of the reef architecture (De Gelder et al. 2022 Fig. 4 might give some inspiration, no need to cite that of course, and apologies for the self-promotion)

We appreciate the inspiration. Each cross section has been expanded to better utilize space.

Fig 8: I think it would be nice to have unconformities clearly marked in the lithology as well, both as a visual aid and to help with the results section. For instance in Fig. 8a, it would be nice to see a separation in the lithology for the Baseline model, where you jump from an ~80 ka forereef to a Holocene forereef. Another thing for this figure is that the lithology weirdly stops at the bottom for some cases (white area), I think this should be an easy cosmetic fix. Is it supposed to be basement?

Agreed. We have gone through and have marked the unconformities within Figure 8 for better readability. We have also added the basement to the bottom of each log for continuity.

Another, more important thing about this figure is that I think it would be worth wile to add the stratigraphy at Well C towards the end of MIS 5e, i.e. before a (large?) portion of the stratigraphy was removed by erosion. Was there any reef flat or beach deposits above the reef crest deposits? That would make for an interesting comparison with the data. Also, maybe there were some beach deposits in the Baseline or G2A5 scenarios at the end of MIS 5e, which would be worth comparing with the observed stratigraphy?

We have added the MIS 5e stratigraphy for Full, G2A5 and Base to Fig. 10 at each scenarios' peak. We choose to explore the peak lithology at different timesteps because this allows us to have a better comparison to potential preservation. This is then further discussed in the text as suggested by your previous comment.